# DEMOCRATIZING EVALUATION BY ∞-BENCHMARKS: SAMPLE-LEVEL HETEROGENEOUS TESTING OVER ARBITRARY CAPABILITIES

## ABSTRACT

Traditional fixed test datasets fall short in quantifying the open-ended potential of foundation models. In this work, we propose ∞-benchmarks, a new testing paradigm that combines individual evaluation datasets into a single, uniform, ever-expanding sample pool from which custom evaluations can be flexibly generated. ∞-benchmarks allows users to dynamically select a collection of sample-level evaluations that correspond to specific capabilities. By aggregating and reusing samples across various test sets, it enables the assessment of diverse capabilities beyond those covered by the original test sets, while mitigating overfitting and dataset bias. Most importantly, it frames model evaluation as a collective process of aggregation and selection of sample-level tests. The shift from multi-task benchmarks to ∞-benchmarks introduces two key challenges: (1) *heterogeneity* and (2) *incompleteness*. *Heterogeneity* refers to aggregating diverse metrics, including binary, numeric, and ordinal data, while *incompleteness* describes comparing models evaluated on different subsets of testing data. To address these challenges, we explore algorithms to aggregate sparse, unequal measurements into reliable model scores. Our aggregation algorithm ensures identifiability (asymptotically recovering ground-truth scores) and rapid convergence, enabling accurate model comparisons with relatively little data. Our algorithm recovers ground-truth rankings with high correlations when compared to standard aggregation on homogeneous metrics, even with up to 95% of measurements missing. This approach reduces evaluation cost by up to $20\times$ with little to no change on model rankings. We introduce ∞-LLMBench for language models and ∞-LMMBench for vision-language models, unifying evaluations across diverse test-beds in these domains, and showcasing targeted testing of models over a wide-range of capabilities. Overall, we present the first large-scale ∞-benchmarks for lifelong, efficient evaluation of foundation models, which can aggregate over open-ended heterogeneous sample-level testing to evolve alongside the rapid development of these foundation models.

## 1 INTRODUCTION

Machine learning has arrived in the post-dataset era[1]. With the rapidly growing range of zero-shot capabilities of foundation models, the focus of model evaluation has moved beyond singular, dataset-specific performance measurements obtained by splitting a fixed collection of data into training and test sets. Instead, foundation models are employed as general knowledge and reasoning engines across the broad suite of domains for which they prove to be useful. There is consequently a pressing need to characterize their open-ended set of capabilities across various metrics in zero-shot settings (Ge et al., 2024). Traditional static benchmarks, however, which test generalization on fixed test splits, are unable to probe the ever-evolving set of capabilities of foundation models. This raises an important question: *How can benchmarking adapt to measure an open-ended set of capabilities?*

We propose a solution based on dynamic sample-level evaluation, which we call ∞-**benchmarks**, where test sets for particular capabilities are generated ad hoc from a large pool of individual annotated data samples. These sample-level evaluations act as atomic units of measurement that can be combined into an exponential variety of aggregations. Due to this flexibility, the sample pool and corresponding annotation metrics can be continuously updated to include new evaluations. Additionally, this can reduce *dataset bias*—systematic quirks in the data arising from the acquisition

---

[1]From a talk by Alexei Efros at ICML 2020

procedures used during dataset collection (Torralba & Efros, 2011; Liu & He, 2024). By combining samples across test sets, ∞-benchmarks can better capture real-world diversity (Ni et al., 2024).

The most important feature of ∞-benchmarks is their ability to efficiently democratize evaluation. Unlike traditional benchmarks, typically created by individual groups arbitrarily deciding on specific data collection and evaluation procedures (Dhar & Shamir, 2021), ∞-benchmarks allow the integration of test sets from many diverse sources reflecting a wide range of perspectives, use-cases, and objectives. This flexibility allows different interest groups with varying needs to collaboratively define their own evaluations selecting the most appropriate combination of tests to suit their specific requirements. Moreover, the design of ∞-benchmarks challenges the dominant approach of chasing single benchmark scores in favour of a plurality of rankings and dynamic, multi-faceted evaluation.

**Challenges in ∞-Benchmarks.** To build effective ∞-benchmarks, we must address two main challenges: **(a)** *Heterogeneity* and **(b)** *Incompleteness*. *Heterogeneity* refers to aggregating samples over an ever-expanding set of metrics, which span different measurement types—including binary, numeric, and ordinal data. This diversity makes it difficult to standardize comparisons across different models. *Incompleteness*, on the other hand, arises from models being evaluated on different, unequal subsets of testing data, rendering direct aggregation unfair. Traditional benchmarks typically use a multi-task benchmarking setting, where each component benchmark still evaluates models over an equal, fixed sample set across a homogeneous metric, completely sidestepping both these issues.

Solution and Theoretical Guarantees. To tackle these challenges, we apply social choice theory, viewing samples as voters expressing preferences among models. By converting all measurements into ordinal rankings, we leverage well-established principles to develop a sound model for aggregating over diverse and incomplete data. We assume a random utility model, generated by the Plackett-Luce framework, which provides guarantees on recovering ground-truth model utility scores from input samples. This approach ensures that our model rankings are both theoretically sound and practical, with rapid convergence guarantees enabling accurate rankings from relatively small amounts of data.

Empirical Validation. We develop two instantiations of ∞-benchmarks: ∞-LLMBench for language models and ∞-LMMBench for multimodal models. These benchmarks unify evaluations across their respective domains by aggregating data from diverse sources, from arena-style human preference data (Chiang et al., 2024; Lu et al., 2024b) to heterogeneous multi-task leaderboards (Beeching et al., 2023; Liang et al., 2023; Zhang et al., 2024b; CRFM, 2024). Our empirical results demonstrate that the Plackett-Luce model (Plackett, 1975; Luce, 1959) is a good fit for aggregating real-world benchmarks, showing high correlations with ground-truth rankings over homogeneous leaderboards. Importantly, we demonstrate that this strong correlation holds even when up to 95% of the data is missing. Conversely, this robustness allows us to reduce costs by $20\times$ with little loss in performance. We observe that the simple strategy of randomly selecting a subset of samples achieves comparable performance to more sophisticated sample selection strategies. Finally, we compare Plackett-Luce rankings with widely adopted ranking metrics like ELO Elo (1967) and Bradley-Terry (Bradley & Terry, 1952) and outperform them on overall accuracy and robustness to missing information.

Personalized Aggregation. Consider this scenario: *you are a scientist in a Biochemistry lab and require an LLM to assist with designing experiments related to antibodies.* ∞-benchmarks allow users to input a query, "`biochemistry`"/"`antibodies`", and receive dynamically constructed benchmarks. This benchmark ranks models based on their performance on *this specific capability*. While optimal selection of personalized capability sets is an emerging research field, we provide a proof of concept by categorizing capabilities into *tasks* (e.g., reading comprehension) and *concepts* (e.g., Clostridium Bacteria), and showcasing targeted capability evaluation and model rankings.

In essence, ∞-benchmarks are a democratized, open-source collection of diverse evaluation samples and model measurements, with detailed metadata. Users can conduct semantic searches and apply structured query filters to dynamically generate a benchmark tailored to their specific use case. Sample-level model measurements can be instantly aggregated, producing personalized rankings.

## 2 AGGREGATION IN ∞-BENCHMARKS: THEORY AND PRACTICE

We view aggregating sparse ordinal preferences over models through a computational social choice lens (Brandt et al., 2016)—samples are voters, models are candidates, and the aggregation algorithm is the voting mechanism. Using established methods, we aggregate ordinal comparisons with partial data to produce a global ranking and analyze properties of this resultant ranking.

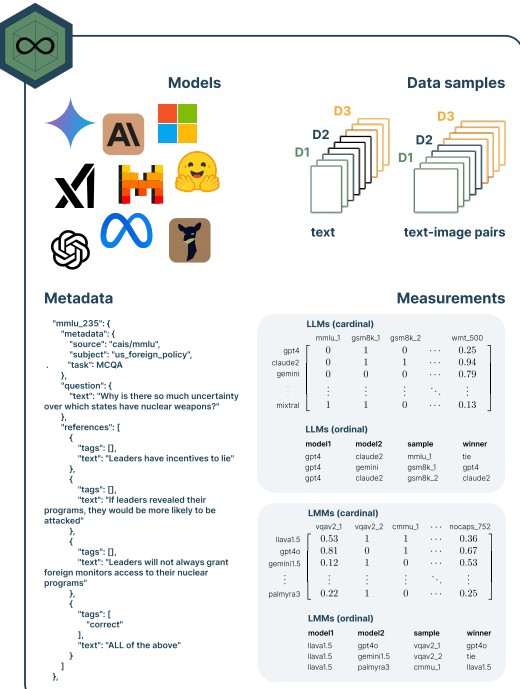 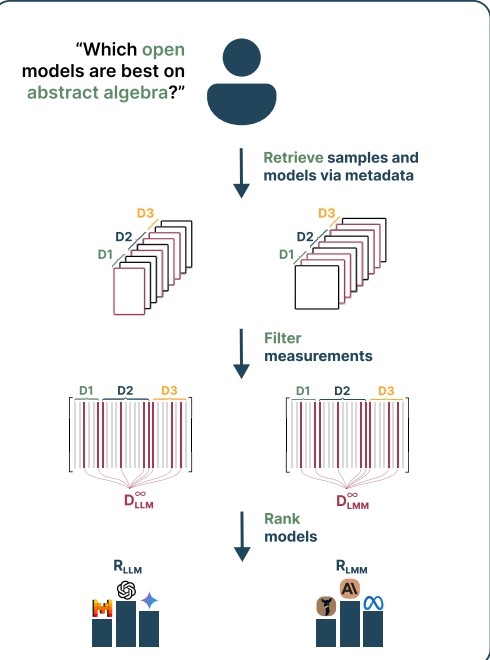

Figure 1: **The ∞-Benchmark Framework.** *(left)*: an ∞-Benchmark comprises a set of models, a pool of data samples spanning multiple test datasets, metadata describing models and data samples, and a collection of heterogeneous, sample-level measurements. *(right)*: the user formulates a query that reflects the desired model capability through a mix of structured metadata filters and semantic search. Selected models are then ranked on a subset of data samples that meet the specified criteria.

## 2.1 THEORETICAL FOUNDATIONS: WHY THIS WORKS?

We begin by postulating a ground-truth statistical model generating the data, which is converted into ordinal comparisons $(\mathcal{S})$[2]. Specifically, we use a random-utility model (Thurstone, 1927), where each model $f_i$ is associated with a utility distribution $\mathcal{U}_{f_i}$. Preferences between models $f_i$ and $f_j$ are based on comparing sampled utilities, i.e., $f_i \prec f_j := u(f_i) < u(f_j)$, where $u_f \sim \mathcal{U}_f$. Since computing maximum likelihood estimates over general random-utility models is computationally hard (Xia, 2019), we focus on the Plackett–Luce model (Plackett, 1975; Luce, 1977), the only known exception that allows for tractable maximum likelihood estimates (MLE).

**P1. Identifiability.** We first ask: *Are the utility distributions across models $\mathcal{U}_{f_i} \forall f_i$ recoverable?* The Plackett-Luce model allows identifying the utility distribution (up to arbitrary additive constant) if all models are compared via a directed path (Xia, 2019; Hunter, 2004)[3]. Consistency and asymptotic normality hold under specific assumptions about the comparison graph (Han & Xu, 2023).

**P2. Sample-Efficient Convergence from Sparse Data.** Identifiability is asymptotic, but we also ask: *How sample-efficient are algorithms for recovering the utility distribution?* With partial rankings of size $k$, the MLE is surprisingly sample efficient while being minmax-optimal (Hajek et al., 2014; Maystre & Grossglauser, 2015). Specifically, sampling $k$ model comparisons from the model set $|\mathcal{F}|$ independently and uniformly at random for $|\mathcal{D}|$ samples induces an expander graph with high probability, which provides guarantees on sample-efficiency of recovery, with $|\mathcal{D}| = \Omega(|\mathcal{F}|)/k$ samples being necessary, and $|\mathcal{D}| = \Omega(|\mathcal{F}| \log |\mathcal{F}|)/k$ samples being sufficient. Efficient algorithms like those in Agarwal et al. (2018); Maystre & Grossglauser (2015) achieve these bounds. Rank-breaking techniques, used in our empirical evaluation, also offer near-optimal solutions (Soufiani et al., 2014).

**P3. Active Aggregation.** In ∞-benchmarks, we can strategically select model comparisons, by framing selection as an online multi-armed bandit problem. One can provide significantly more

---

[2]This contrasts with Zhang & Hardt (2024), who view aggregation as classical voting, analysing tradeoffs in aggregating voter preferences rather than uncover an underlying ranking.

[3]Recall that using the reference model $f_{\text{base}}$ removes the additive ambiguity.

sample efficient convergence with PAC guarantees (Szörényi et al., 2015; Saha & Gopalan, 2019; Ren et al., 2018), significantly outperforming random comparisons (Maystre & Grossglauser, 2017).

**P4. Social Properties.** The Plackett-Luce model ensures computational efficiency and recoverability of the underlying ranking. However, to design democratic systems for decision-making, it is essential to also have fair aggregation. However, ensuring fairness involves tradeoffs (Zhang & Hardt, 2024) because different notions of fairness often conflict, and agents may have differing, even opposing preferences (Garman & Kamien, 1968; Arrow, 1950; Benoît, 2000). We, however, can state that Plackett-Luce model is procedurally fair (List, 2022) (Section 2.2), i.e. it satisfies:

- Anonymity. All voters (samples) are treated equally, ensuring the system does not rely on a single vote. The rankings unchanged even if the input sample set is permuted.
- Neutrality. The ranking is invariant to the identities of the models, ensuring fairness among alternatives. This means permuting the models similarly permutes the new ranking.
- Independence from Irrelevant Alternatives (IIA). The relative ranking of two models is unaffected by other alternatives in a given sample, as guaranteed by Luce's axiom of choice (Luce, 1959). This provides grounding for incomplete model evaluations.

### 2.2 TRANSLATING THEORY TO PRACTICE: EMPIRICAL VALIDATION

We now empirically validate our framework, aiming to show that: (i) the Plackett-Luce model fits real-world data well, (ii) our aggregation method is sample-efficient, and (iii) it handles high levels of incompleteness. Furthermore, we discuss practical strategies for reducing evaluation costs in Sec. 2.3. Below, we describe our setup and address these points.

#### 2.2.1 SETUP

**Benchmarks.** We conduct experiments using four popular leaderboards with established ground truth model rankings: HELM (Liang et al., 2023) and Open-LLM Leaderboard (Beeching et al., 2023) for LLMs, and VHELM (CRFM, 2024) and LMMs-Eval (Zhang et al., 2024b) for LMMs. We fix our sample pool as all samples from the constituent datasets of a given leaderboard and compare rankings obtained by our aggregation strategy. These leaderboards evaluate foundation models across varied tasks with different metrics, serving as good indicators of real-world performance.

**Methods.** We evaluate three model ranking methods:

(i) Elo Score: (Elo, 1967) A competitive game rating system adapted to rank models through pairwise comparisons, adjusting scores based on wins or losses to reflect win-rate reliability.

(ii) LMArena Ranking: (Chiang et al., 2024) A method for LLM ranking using the Bradley-Terry model (Bradley & Terry, 1952), which estimates model rankings through Maximum Likelihood Estimation (MLE) based on pairwise comparisons using an underlying ELO model.

(iii) Our Method: Our approach leverages the Plackett-Luce model (Maystre & Grossglauser, 2015) to aggregate pairwise comparisons using partial rank breaking (Soufiani et al., 2014).

**Metrics.** We compare the rankings generated by each method to the ground-truth from the leaderboards using Kendall's $\tau$, a standard correlation metric for rankings. Each method is tested thrice, and we report the mean and variance. We additionally check that the top-$k$ models are reliably recovered.

#### 2.2.2 P1: IS PLACKETT-LUCE A GOOD FIT FOR REAL-WORLD DATA?

| Metric | HELM | Open-LLM Leaderboard | LMMs-Eval | VHELM |
|---|---|---|---|---|
| Elo Score | 0.347 ± 0.132 | 0.213 ± 0.065 | 0.363 ± 0.109 | 0.639 ± 0.024 |
| LMArena Ranking | 0.952 ± 0.001 | 0.969 ± 0.000 | 0.473 ± 0.000 | 0.697 ± 0.000 |
| Our Method | **0.977 ± 0.001** | **0.997 ± 0.000** | **0.670 ± 0.000** | **0.827 ± 0.000** |

Table 1: **Kendall's $\tau$ correlations of aggregation algorithms** along with ground-truth rankings. Results show improvements over ELO and LMArena rankings, with notable correlation boosts on $\infty$-LMMBench leaderboards, including LMMs-Eval (41.65%) and VHELM (14.63%).

Q1. Is it a good fit? We assess whether the Plackett-Luce model performs well on large-scale benchmark data by comparing our aggregation algorithm's rankings to the leaderboard rankings. As shown

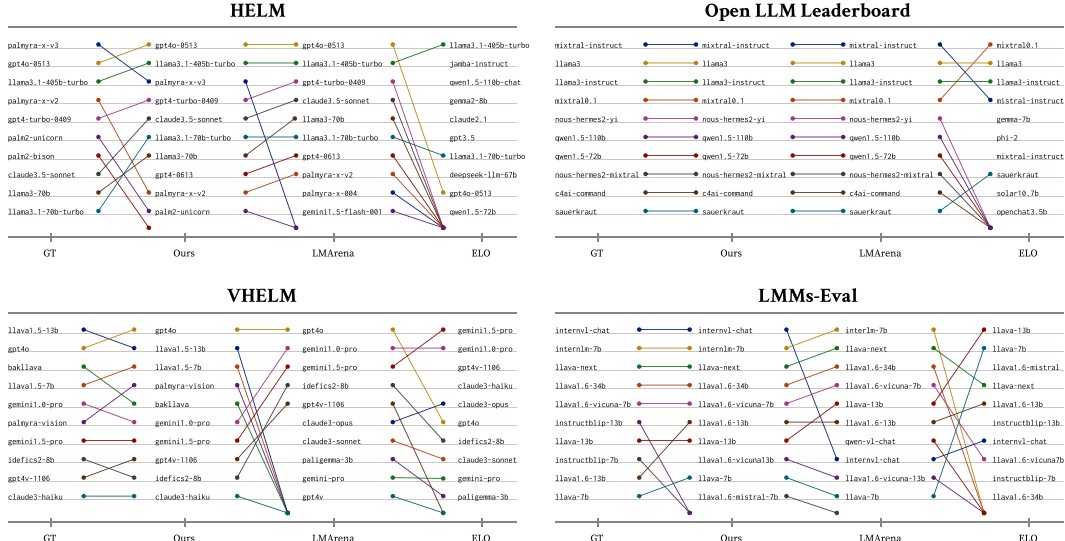

Figure 2: **Top-10 model ranking changes across different aggregation methods.** A progressive degradation in ranking accuracy is observed from ground truth (`GT`) to our method (`Ours`), LMArena Scores (`LMArena`), and Elo scores (`ELO`). Comparisons are shown for $\infty$-LLMBench (top) and $\infty$-LMMBench (bottom). Our method preserves the ranking of the top-10 models.

in Table 1, our algorithm achieves a high positive Kendall's $\tau$, indicating strong alignment with the ground truth rankings.

**Q2. Is it better than current metrics?** In addition to evaluating fit, we also compare our method to popular algorithms like Elo and LMArena. Table 1 shows that our algorithm consistently outperforms these methods, demonstrating its superior performance for large real-world datasets.

**Q3. Are the top-k models preserved?** For practitioners, the critical concern is whether the top models are ranked correctly. Figure 2 shows that our algorithm effectively preserves the top-10 model rankings compared to ground truth, while outperforming state-of-the-art methods in maintaining accurate top-k rankings.

**Conclusion.** The Plackett-Luce model fits real-world data well, outperforming other methods in both overall Kendall's $\tau$ and top-10 model rankings, making it empirically effective for large-scale benchmarks. The underlying reason is that we avoid the limitations of Elo-based methods, which rely on assumptions that do not apply to foundation models (Boubdir et al., 2023).

### 2.2.3 P2: SURPRISING SAMPLE EFFICIENCY AND HANDLING INCOMPLETE RANKINGS

We now empirically test the sample efficiency and robustness to incomplete data of our framework.

**Q1. Is Our Algorithm Sample-Efficient?** We systematically reduce the number of samples and re-rank the models using various methods, calculating Kendall's $\tau$ for each. Missing data is simulated from 0% to 99%, with 10% intervals until 90%, followed by 1% increments. As shown in Fig. 3, our method maintains stable performance even with up to 95% fewer samples, demonstrating that it can achieve accurate rankings with far fewer data points—up to 20x less than current benchmarks.

**Q2. Can our Algorithm Aggregate Highly Sparse Rankings?** We evaluate the method's ability to handle highly incomplete data by removing model comparisons from the samples and re-ranking the models. We randomly remove a fraction of model measurements from each sample and re-rank using various aggregation methods. Again, we simulate data removal from 0% to 99%, as increments as before. As shown in Fig. 3, our method performs well even with 95% fewer model comparisons, proving it can recover accurate rankings with highly sparse data, crucial for $\infty$-benchmarks where models are evaluated on different samples.

**Conclusion.** Our algorithm provides significant sample efficiency, maintaining accurate rankings with 20x fewer data points, and is robust to highly sparse input rankings.

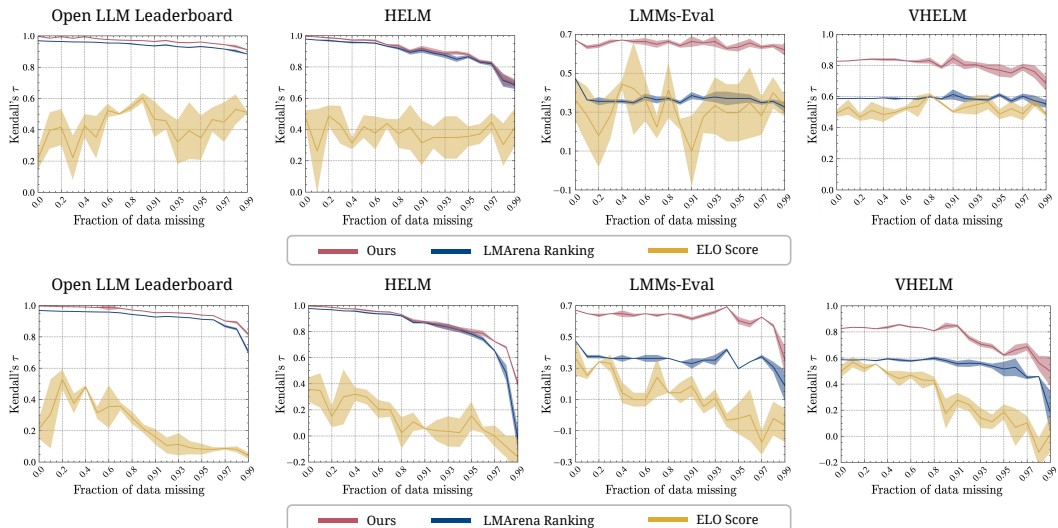

Figure 3: *(top) Sample-efficient convergence and (bottom) Sparsity of k.* Kendall $\tau$ between ground-truth ranking and different ranking methods as data is removed for re-ranking and as sparse rankings are aggregated with model measurements removed. Methods typically remain robust to missing data, with Plackett-Luce consistently achieving higher correlation, even with 95% measurements missing.

## 2.3 P3: ACTIVE SAMPLING IMPROVES DATA AGGREGATION EFFICIENCY

We now explore methods to enhance data aggregation beyond random sampling. We conduct experiments to identify key insights for improving sample efficiency.

**Setup.** We leverage the sample-level design of $\infty$-Benchmarks to inspect the distribution of samples in standard benchmarks. We investigate efficient model evaluation strategies by selecting a small subset of the total pool—we aggregate model accuracies to identify easy and difficult data samples.

Insight 1: Many Samples Provide No Signal. The histograms in Fig. 4 show that a large portion of samples result in identical model scores (all 0s or all 1s), resulting in ties when converted to ordinal ranking and contributing no useful information for model comparison. Excluding these samples could reduce dataset size by up to 50% for benchmarks like Open LLM Leaderboard and LMMs-Eval(comprising close to 150K samples which no model can answer correctly).

Insight 2: Efficient Sampling from Central Bins. By analyzing rank correlation (Kendall's $\tau$) in Fig. 4, we found that sampling from central bins of the data histogram—where models differ in evaluation performance—maintains a higher rank correlation than the edge bins, even with fewer than 400 data points. This indicates that effective sampling can be achieved in $\infty$-benchmarks. While prior studies suggest sampling informative instances (Vivek et al., 2024; Perlitz et al., 2024), others, like (Prabhu et al., 2024), show random sampling can yield strong results.

Insight 3: Random Sampling Matches Informative Sampling. Comparing informative sampling (based on the fraction of models solving a data instance) with random sampling, we found no significant difference in sample efficiency (Fig. 4). This suggests that random sampling is an equally effective and simpler approach for reducing benchmarking costs.

**Conclusion.** Benchmarking large models is resource-intensive, but we demonstrate that excluding low-signal data and relying on random sampling can significantly reduce costs without compromising accuracy. This strategy is effective for large-scale, sample-wise evaluations in $\infty$-benchmarks.

## 3 $\infty$-LLMBENCH & $\infty$-LMMBENCH: CREATION & CAPABILITY QUERYING

After evaluating the robustness of our aggregation method across incomplete and heterogeneous measurements, we present the overall system applied to two large-scale, real-world $\infty$-benchmarks for foundation models: LLMs and LMMs. We first outline how these benchmarks were created, then explain how arbitrary capabilities are tested on them, and highlight key insights gained.

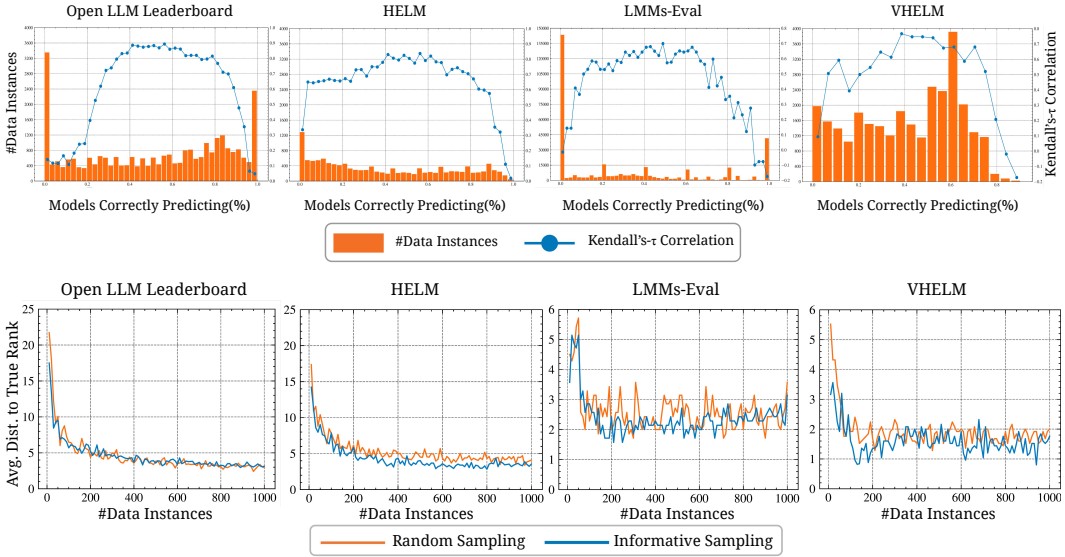

Figure 4: *(top)* Histogram of data instances showing percentage of models that correctly solve them. Most instances in Open-LLM-Leaderboard and LMMs-Eval are either too difficult (no models solve them) or too easy (all models solve them). For each bin, we compute model rankings based on instances in that bin and plot the Kendall-$\tau$ correlation with the global ranking. *(bottom)* Average rank difference between actual and estimated ranks across models. The random strategy selects instances uniformly, while the informative strategy prioritizes instances with maximum model measurement entropy. Both strategies perform similarly, justifying our choice of using random selection strategy.

## 3.1 CREATION OF ∞-LLMBENCH & ∞-LMMBENCH

### 3.1.1 ∞-LLMBENCH

**Data Pool $\mathcal{D}$.** For ∞-LLMBench (Tab. 3), we source data from Open LLM Leaderboard (Beeching et al., 2023), HELM (Liang et al., 2023), and LMArena (Chiang et al., 2024). Open LLM Leaderboard and HELM aggregate several individual benchmarks (*e.g.*, MMLU (Hendrycks et al., 2021a), HellaSwag (Zellers et al., 2019)), while LMArena uses pairwise model comparisons based on user-generated prompts, with user votes determining the superior model. Metrics which are converted to samplewise ordinal rankings here include F1-Scores, Exact Matches (EM), Quasi-Exact Matches (QEM) for binary measurements, and pairwise preferences from LMArena for ordinal measurements.

**Models $\mathcal{F}$.** For ∞-LLMBench, we use 100 most downloaded models from Open-LLM-Leaderboard and 54 from HELM, including both proprietary models like GPT-4o (OpenAI, 2024) and open-weight ones like LLaMA-3 (Meta, 2024). A full list of evaluated models is provided in Appx. D.

### 3.1.2 ∞-LMMBENCH

**Data Pool $\mathcal{D}$.** For ∞-LMMBench (Tab. 4), data is sourced from VHELM, LMMs-Eval, and WildVisionArena. Similar to ∞-LLMBench, VHELM and LMMs-Eval aggregate individual datasets like MMMU (Yue et al., 2024) and VQAv2 (Goyal et al., 2017), while WildVisionArena uses pairwise tests for LMMs through image-based chats. We convert a diverse set of metrics to samplewise rankings, from binary metrics like EM, QEM, to real-valued scores like ROUGE (Lin, 2004), Perception (P) and Cognition (C) scores from MME (Fu et al., 2023). We additionally combine pairwise comparisons from WildVisionArena with LLM-As-A-Judge preferences using Prometheus-2 (Kim et al., 2024), which correlates highly with human judgment, with preference comparisons are sampled randomly from LMMs-Eval while avoiding overlap with cardinal measurements

**Models $\mathcal{F}$.** For ∞-LMMBench, we use 14 models from LMMs-Eval (Zhang et al., 2024b) and 25 models from VHELM (CRFM, 2024), including open-weight models like LLaVA (Liu et al., 2023a) and proprietary models like Gemini Pro Vision (Team et al., 2023). A complete list of evaluated models is provided in Appx. D.

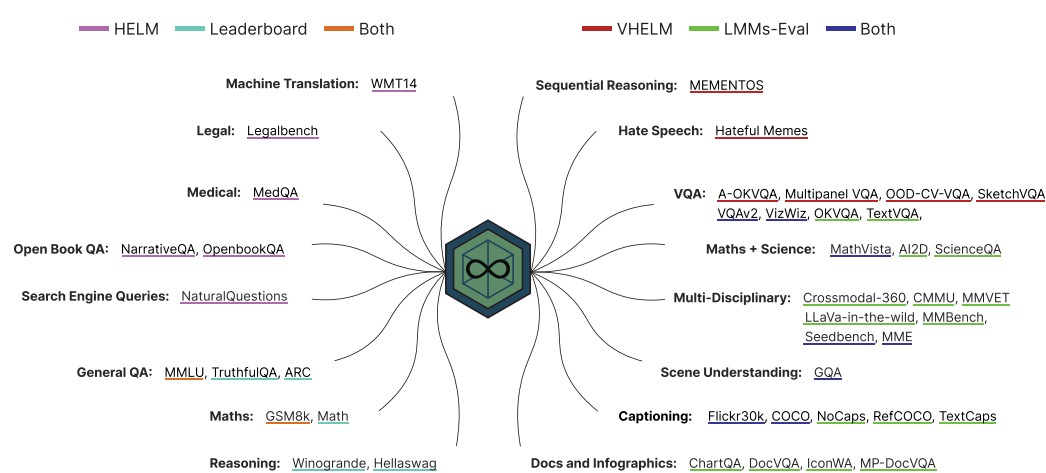

Figure 5: Constituent datasets of ∞-**LLMBench (left)** and ∞-**LMMBench (right)** along with task metadata. We provide details including task type, metric, and license about each dataset in Appx. C.

## 3.2 CAPABILITIES AND CONCEPT PROBING

Here, we present empirical results on generating arbitrary test sets and rankings. Our goal is to enable users to make targeted queries within ∞-benchmarks, helping them identify the best LLMs and LMMs for their specific needs. To achieve this, we extend our system with a flexible mechanism for personalized aggregation, allowing users to (1) retrieve relevant data instances through semantic search, and (2) dynamically generate rankings based on the retrieved samples.

**Setup.** The user submits a query, and we retrieve relevant data samples using semantic search. This *concept querying mechanism* provides a personalized comparison of foundation model capabilities. We use two querying mechanisms: (i) Semantic search, where we use `all-MiniLM-L6-v2` (Reimers & Gurevych, 2019) for language tasks and `SigLIP-B16` (Zhai et al., 2023) for vision-language tasks, employing cosine similarity for retrieval. We retrieve top-k samples for a given concept with a well-tuned cut-off similarity score of 0.3 and 0.7 for ∞-LLMBench and ∞-LMMBench respectively. (ii) Metadata search: We search metadata to match querying. With this, we gather representative samples for the query, and aggregate the ordinal model rankings per sample using the Plackett-Luce model to produce final model rankings, for that particular query.

**Concepts Tested.** We curated a diverse set of 50 concepts to test the breadth and versatility of our ∞-benchmarks, ranging from domain-specific knowledge, such as the Coriolis Effect, to broader academic disciplines like Neuroscience, and everyday consumer goods like the Apple iPad. We showcase 6 of them in the main paper, and present the rest in Appx. E.

### 3.2.1 RESULTS & INSIGHTS

We present the results from concept querying in Figure 6 and summarize our insights below:

Insight 1. Are the retrieved datasets accurate? Two expert annotators manually reviewed and filtered out incorrect matches [4]. To evaluate the quality of the retrieved samples, we report average precision (AP) scores for a random subset of queried concepts in Fig. 6, with a full list of scores in Appx. E. Aggregating over all tested concepts in Table 2, our mAP over the concepts is 0.84 and 0.73 for ∞-LLMBench and ∞-LMMBench respectively, demonstrating that we can reliably retrieve samples that match the intended capabilities, although there is substantial scope for improvement in some cases (like neuroscience in ∞-LMMBench). Note that the retrieval mechanism is expected to only improve with better foundation models and more sophisticated querying mechanisms are integrated in ∞-benchmarks.

Insight 2. Do models perform differently across queries? A key check is to verify whether models perform distinctly across different capability queries. If the results are similar regardless of the query, fine-grained querying may be less useful, as the top model from a generic leaderboard could be a

---

[4]The inter-annotator agreement, measured by Cohen's Kappa, is shown in Table 2, with high values of 0.793 and 0.912 indicating strong consistency between annotators.

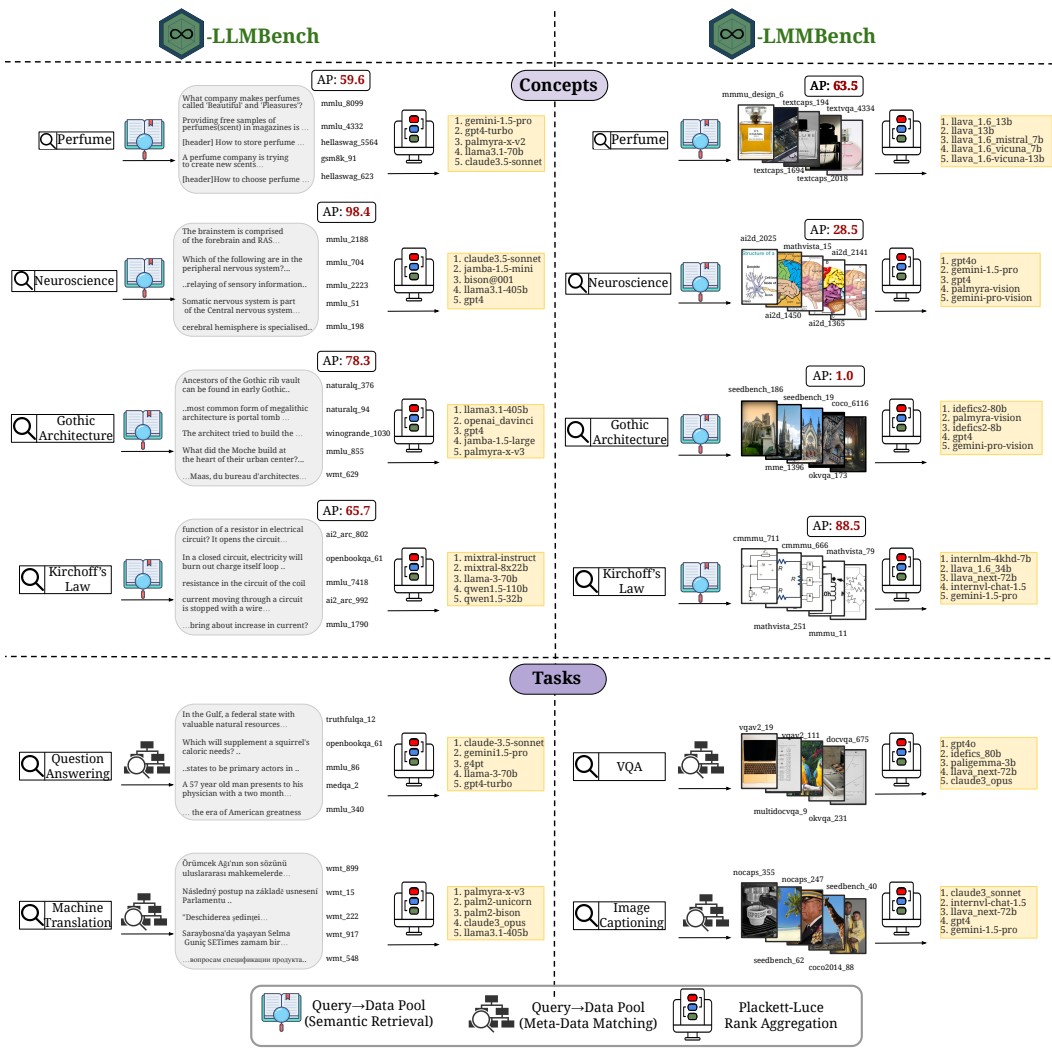

Figure 6: **Capability Probing(Qualitative):** We provide six sample retrieval results for a set of queries covering a diverse set of topics and report the top-5 models for each query.

| Benchmark | #Concepts | Cohen-$\kappa$ | mAP | CMC@1 | CMC@10 |
|---|---|---|---|---|---|
| ∞-LLMBench | 40 | 0.793 | 0.8462 | 0.95 | 1.0 |
| ∞-LMMBench | 50 | 0.912 | 0.7337 | 0.94 | 0.96 |

Table 2: **Capability Probing(Quantitative):** We provide a summary of the number of concepts curated for capability probing, along with the (high) inter-annotator agreement and retrieval metrics.

good candidate for any specific capability, as is common practice currently. However, we observe in Figure 6 that very different models perform well on different domains, concepts. This enables ∞-benchmarks to scalably return good candidate models, customized for arbitrary user queries.

## 4 RELATED WORKS

While recent benchmarks have tested broad capabilities of foundation models, viewing benchmarking as a science ((Hardt & Recht, 2022)) is understudied. We provide a short overview of recent efforts, highlighting the intersectional nature of our work We include a detailed version in Appx. B.

**Multi-task Benchmarks.** Multi-task leaderboards, e.g., GLUE (Wang et al., 2019b), Super-GLUE (Wang et al., 2019a), and BigBench (Srivastava et al., 2023), are standard for evaluating

foundation models across tasks. However, concerns about dataset selection and saturation have emerged (Ethayarajh & Jurafsky, 2020; Dehghani et al., 2021; Liu & He, 2024). Our ∞-benchmarks address these by enabling extensive reuse of samples, avoiding task selection bias (Torralba & Efros, 2011; Dominguez-Olmedo et al., 2024), and supporting open-ended evaluations through querying for diverse concepts across a broad range of input metrics and incomplete set of model comparisons.

**On Aggregation across Benchmarks.** Traditional benchmarks use arithmetic mean for task aggregation (Beeching et al., 2023) which can distort rankings (Benavoli et al., 2016; Zhang & Hardt, 2024; Colombo et al., 2022a) and unusually depend on outliers (Agarwal et al., 2021) and missing scores (Himmi et al., 2023). Inspired by non-parametric statistics and social choice theory, we employ ordinal rankings and the Plackett-Luce model (Plackett, 1975) for task aggregation, which is robust to irrelevant alternatives and outliers, providing more accurate and efficient evaluations.

**Efficient Evaluation and Democratization.** As benchmarks grow, so do inference costs, leading to compressed subsets (Varshney et al., 2022; Polo et al., 2024; Vivek et al., 2024; Zhao et al., 2024; Perlitz et al., 2024) and evolving lifelong benchmarks (Prabhu et al., 2024). Our approach, for the first time, enables past work to handle incomplete data and ordinal rankings. Further, by allowing diverse contributors to add samples and preferences, along with arbitrary queries, we hope ∞-benchmarks can bemore inclusive than traditional benchmarks dominated by well-funded labs following recent progress (Pistilli et al., 2024; Pouget et al., 2024; Nguyen et al., 2024; Luccioni & Rolnick, 2023).

## 5 CONCLUSIONS AND OPEN PROBLEMS

This work tackled scalable benchmarking of arbitrary capabilities of foundation models, requiring a shift from traditional fixed training and test splits, by introducing ∞-benchmarks, a lifelong benchmarking framework for foundation models. Our open-source, democratized benchmarking methodology allows diverse evaluation samples and model measurements with detailed metadata. This affords creating customized benchmarks and testing arbitrary capabilities, including using semantic and structured searches. We provide a principled aggregation mechanism, that is both theoretically grounded and empirically validated to be robust to incomplete data and heterogenous measurements across evaluations. We demonstrate the utility of ∞-benchmarks in two domains: ∞-LLMBench and ∞-LMMBench, showing how dynamic probing reveals new insights into model performance on specific tasks, domains, or concepts. This combination of theoretical rigour, empirical results, and practical flexibility makes ∞-benchmarks a valuable tool for comprehensively evaluating foundation models. we provide some promising directions for improvement below:

1. Testing Limits and Scaling Up ∞-Benchmarks: Currently, our prototype demonstrates the core methodology of ∞-benchmarks, with less than 100K samples in ∞-LLMBench and under 1M in ∞-LMMBench. These pools can be greatly expanded and diversified by expanding to incorporating *all existing* LLM and LMM benchmarks. Our retrieval mechanisms are designed to scale efficiently as the test pool grows in size and diversity.

2. Exploring Aggregation Algorithms from Computational Social Choice: While we currently use the Plackett-Luce model for aggregating diverse measurements, there exist other algorithms from computational social choice theory with different trade-offs. A comprehensive evaluation of these alternatives could offer new insight for aggregating model performance.

3. Structured Querying and Enhanced Retrieval: One can improve retrieval by better querying mechanisms using models like ColBERT (Khattab & Zaharia, 2020) and ColPALI (Faysse et al., 2024) and optimization using DSPy (Khattab et al., 2023). A particularly interesting direction is allowing compositional queries, where users combine multiple queries to test behaviour in foundation models, similar to works like ConceptMix (Wu et al., 2024) and SkillMix (Yu et al., 2023).

4. On the Limits of Capability Probing: While we currently allow broad, open-ended inputs to probe capabilities, some are easier to assess than others (Madvil et al., 2023; Li et al., 2024b). As foundation models become more generalizable, a thorough analysis identifying which capabilities can be *easily, reliably evaluated*, which are *possible to evaluate but challenging*, and which are in principle *"impossible to evaluate"* is needed—this will help improve benchmarking effectiveness.

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

# Part I

# Appendix

# Table of Contents

# A $\infty$-BENCHMARKS: FORMULATION

At the heart of $\infty$-benchmarking is the idea to homogenize performance evaluation across benchmarks by replacing their benchmark-specific *metrics* with *rankings*. Importantly, this can be done at the level of individual data samples. In the following, we describe the process of construction and evaluation in detail, together with specific mathematical guarantees.

## A.1 COMPONENTS

The goal of building an $\infty$-benchmark ($\mathcal{B}_\infty$) from a growing set of benchmarks $(\mathcal{B}_k)_{k=1}^N$ is to evaluate a collection of models ($\mathcal{M}$) using an ever-growing test pool of data instances ($\mathcal{D}$) which may be annotated with additional meta-data specifying the capabilities ($\mathcal{C}$) tested. To cope with the diversity of data originating from diverse benchmarks, sample-level rankings ($\mathcal{S}$) are created for all data instances in the test pool. We provide a schematic overview of $\infty$-benchmarks in Fig. 1 and describe each component below:

**i) Pool of Data.** $\mathcal{D}=((x_1, y_1), \ldots, (x_n, y_n))$ denotes an ordered collection of test data instances $x_i$ with annotation $y_i$. An example of a data instance $x_k$ is the question *'What was the dominant strain of Flu Virus in 2010? Select among the four choices.'* with the reference answer *'H1N1/09'* represented by $y_k$. In addition, information about capabilities can be provided as meta data for example as a list of keywords such as *'temporal Q&A, pandemics, history, biology, virology, multiple-choice Q&A, etc'.*, beyond the specific dataset it originates from. Typically, the data samples are obtained via pooling from $N$ different benchmarks $(\mathcal{B}_k)_{k=1}^N$ and we refer to the subset of data instances obtained from benchmark $\mathcal{B}_k$ as $\mathcal{D}_k \subseteq \mathcal{D}$.

**ii) Models.** $\mathcal{M}=(f_{base}, f_1, \ldots, f_m)$ is a set of $m$ models, whose capabilities are evaluated with respect to a baseline $f_{base}$. Example of $f_{base}$ is a random model. The original benchmarks $(\mathcal{B}_k)_{k=1}^N$ will likely cover different sets of models $\mathcal{M}_{\mathcal{B}_k} \subseteq \mathcal{M}$ for their evaluations.

**iii) Sample-level Rankings.** For each data instance $(x_j, y_j) \in \mathcal{D}$ a sample level ranking $s_j \in \mathcal{S}$ is created for the subsets of models $\mathcal{M}_j = \mathcal{M} \cap \mathcal{M}_{\mathcal{B}_{k(j)}}$ where $k(j)$ denotes the index of the benchmark from which the data instance $(x_j, y_j)$ was collected. Importantly, sample-level rankings are a function of the metrics used by the different benchmarks that discards any information about the specifics of the metrics. This is the key of our approach to enable the aggregation across heterogeneous evaluation paradigm and metrics. More specifically, $s_j \in \mathcal{S}$ represents an ordinal ranking over the models $\mathcal{M}_j$ for sample $(x_j, y_j)$ represented by a permutation $\sigma_j$ such that $f_{\sigma_j(1)} \succeq \cdots \succeq f_{\sigma_j(m_j)}$ where $m_j = |\mathcal{M}_j|$ is the number of models compared in the $j$-th sample-level ranking. In addition, for each $k$ we distinguish the case $f_{\sigma(k-1)} \succ f_{\sigma(k)}$ if $f_{\sigma(k-1)}$ performs better than $f_{\sigma(k)}$ and $f_{\sigma(k-1)} \sim f_{\sigma(k)}$ in case of indistinguishable performance. Thus, each sample-level ranking $s_j \in \mathcal{S}$ can be uniquely determined by a mapping $\sigma_j : \{1, \ldots, m_j\} \rightarrow \{1, \ldots, m\}$ with $\sigma_j(k)$ providing the index of the model in $\mathcal{M}$ that is on the $k$-th place in the ordering for the $j$-th sample-level ranking and $\pi_j \in \{\succ, \sim\}^{m_j-1}$ defining the corresponding binary sequence of pairwise performance relations.

Ordinal Rankings and Information Loss. Using ordinal measurements leads to information loss, which can hinder downstream aggregation algorithms due to the data processing inequality ((Thomas & Joy, 2012), Section 2.8). This principle states that estimation from manipulated data cannot outperform estimation from the original data. However, cardinal measurements often face calibration issues, even within a single metric (Shah et al., 2014). As a result, in practice, ordinal measurements can paradoxically outperform cardinal ones despite the inherent information loss.

**iv) Capabilities.** To support the selective retrieval of all relevant sample-level rankings in $\mathcal{B}$ based on varying interests of evaluators in different capabilities, it is possible to endow the sample-level rankings with additional *capability* $c \in \mathcal{C}$. Of course, modeling the range of *capabilities* that different evaluators are interested in is a research challenge in itself. Here, we only provide a proof of concept, for which we define two categories of capabilities, including *tasks*, like multiple-choice question answering, captioning, translation, to *concepts* like makeup, dogs, $\pi$, tarantula. The reason behind the broad interpretation is for $\infty$-benchmarks is to test which capabilities can be reliably tested dynamically. Note that since the capability set is open-ended, we do not append capabilities per sample as meta-data, but rather select relevant samples at test-time.

Continual Expansion of $\infty$-Benchmarks. The data instance pool ($\mathcal{D}$), and model names ($\mathcal{M}$) are stored as a table while sample-level testings ($\mathcal{S}$) are stored as a relational database between these two

tables. Construction of a lifelong heterogenous benchmark augments $\mathcal{D}$, $\mathcal{M}$ and $\mathcal{S}$ with three operations: $\mathcal{B}{=}(\mathcal{D}, \mathcal{M}, \mathcal{S}, \text{insert}_\mathcal{D}, \text{insert}_\mathcal{M}, \text{insert}_\mathcal{S})$. Operations $\text{insert}_\mathcal{D}$ and $\text{insert}_\mathcal{M}$ for expanding the data pool are straightforward: add new samples and new models to the corresponding table. The $\text{insert}_\mathcal{S}$ operation adds a new sample-level ranking, each of which corresponds to *one* sample and a *ranking* of models. Additional measurement metadata is saved to enable retrieval over database rows with the same metadata, such as 'BLEU score', or 'exact match'.

## A.2 QUERYING CAPABILITIES FOR PERSONALIZED EVALUATION

To evaluate a given capability, $\infty$-benchmarks take a dynamic approach. First, we randomly select a group of samples from a larger pool that matches the query. Then, we combine these standardized measurements into a final score. This process consists of: (i) Subsample ($\text{retrieve}_\mathcal{D}$), (iii) Aggregate ($\text{Aggregate}_{\mathcal{S},\mathcal{S}}$).

**i) Retrieve (`retrieve`$_\mathcal{D}$).** In this step, the system selects relevant data instances based on a user's query. The query language is flexible and allows retrieving data instances that semantically relate to a specific topic or match certain criteria. The retrieval is implemented through a combination of k-nearest neighbors (kNN) search on dense embeddings using the query as the input and structured queries that take advantage of the unified data schema. We provide extensive empirical analysis to validate the efficacy of this operation.

**iii) Aggregate (`Aggregate`$_{\mathcal{S},\mathcal{D}}$).** We combine the measurements from the retrieved subset of data instances using the random utility modeling approach (Xia, 2019) which defines a joint probability distribution over all measurements assuming statistical independence:

$$p(s_1, \ldots, s_{n_\infty} | \gamma_1, \ldots, \gamma_m) = \prod_{j=1}^{n_\infty} p(s_j = [.]_{(\sigma_j, \pi_j)} | \gamma_1, \ldots, \gamma_m)$$

The Placket-Luce model assumes the following probability model:

$$p\left(s_j = [.]_{(\sigma_j, \pi_j)}\right) = \underbrace{\frac{\gamma_{\sigma_j(1)}}{\sum\limits_{k=1}^{m_j} \gamma_{\sigma_j(k)}}}_{f_{\sigma_j(1)}} \times \underbrace{\frac{\gamma_{\sigma_j(2)}}{\sum\limits_{k=2}^{m_j} \gamma_{\sigma_j(k)}}}_{f_{\sigma_j(2)}} \times \cdots \times \underbrace{\frac{\gamma_{\sigma_j(m_j-1)}}{\gamma_{\sigma_j(m_j-1)} + \gamma_{\sigma_j(m_j)}}}_{f_{\sigma_j(m_j)}}$$

defining one parameter $\gamma_k$ for each model $f_k$ that determines its performance relative to all other models. To aggregate the model performances over all sample-level rankings, we determine the parameters

$$\hat{\gamma}_1, \ldots \hat{\gamma}_m = \operatorname*{argmax}_{(\gamma_1, \ldots \gamma_m) \in \mathbb{R}^m} \log p(s_1, \ldots, s_{n_\infty} | \gamma_1, \ldots, \gamma_m)$$

with maximum likelihood estimation. The global ranking is given by the permutation $\sigma_\infty$ for which $\hat{\gamma}_{\sigma_\infty(1)} > \cdots > \hat{\gamma}_{\sigma_\infty(m)}$. The maximum likelihood condition uniquely determines all performance parameters $\hat{\gamma}_k, k = 1, \ldots, m$ as the likelihood function is strictly concave. The parameters of the Plackett-Luce model is identifiable up to an arbitrary additive constant. Consistency and asymptotic normality can also be shown under certain assumptions about the comparison Graph (Han & Xu, 2023). We refer to the estimated latent variables $\hat{\gamma}_k, k = 1, \ldots, m$ as *model scores* or *preformance parameters* with higher values indicating that a model is more likely to perform better on a randomly picked sample-level ranking than one with lower values. To fix the arbitrary additive constant, we set the score of the baseline model $\hat{\gamma}_{baseline} = 0$ to zero.

## B  RELATED WORKS

**Multi-task Benchmarks as Broad Capability Evaluators.** Multi-task leaderboards have been the standard for benchmarking foundation models that generalize across various situations and solve complex tasks. Examples include GLUE (Wang et al., 2019b), decaNLP (McCann et al., 2018), Super-GLUE (Wang et al., 2019a), BigBench (Srivastava et al., 2023), OpenLLM-Leaderboard (Beeching et al., 2023), CLIP-Benchmark (LAION-AI, 2024), ELEVATOR (Li et al., 2022) and DataComp-Evals (Gadre et al., 2023) as well as massive multitask benchmarks like XTREME (Siddhant et al., 2020) and ExT5 (Aribandi et al., 2021). However, concerns have arisen regarding the limitations of multi-task benchmarks (Bowman & Dahl, 2021). Issues include saturation and subsequent discarding of samples (Liao et al., 2021; Beyer et al., 2021; Ott et al., 2022; Ethayarajh & Jurafsky, 2020), susceptibility to dataset selection (Dehghani et al., 2021), obscuring progress by evaluation metrics (Schaeffer et al., 2023; Colombo et al., 2022b), training on test tasks (Udandarao et al., 2024), and data contamination (Elangovan et al., 2021; Magar & Schwartz, 2022; Golchin & Surdeanu, 2024; Deng et al., 2023; Sainz et al., 2023; Golchin & Surdeanu, 2023; Sainz et al., 2024). $\infty$-benchmarks helps tackle these challenges by enabling the extensive reuse of samples for broader model comparisons, avoiding task selection bias through democratized sourcing of samples, and using ordinal rankings to avoid evaluation minutia. Sample-level evaluations with sparse inputs also allow selective removal of contaminated data from targeted models for fairer comparisons and make it harder to train on all test tasks by supporting open-ended evaluations, compared to leaderboards with fixed test sets.

**On Aggregation across Benchmarks.** Since the current dominant form of benchmark was multi-task benchmarks, the dominant aggregation strategy was arithmetic mean over scores across individual benchmarks. However, mean-scores inherently assumes different scoring metrics are homogeneous, scaled correctly and treats treating tasks of different complexity equally (Mishra & Arunkumar, 2021; Pikuliak & Šimko, 2023). In consequence, simple normalization preprocessing changing rankings (Colombo et al., 2022a), and the rankings nearly entirely dependent on outlier tasks (Agarwal et al., 2021), change rankings even with simple alternate aggregations like geometric/harmonic mean (Shavrina & Malykh, 2021) and including irrelevant alternative models can change statistical significance or even change the ranking entirely (Benavoli et al., 2016; Zhang & Hardt, 2024). Mean-aggregation also has significant failure modes in handling missing scores in benchmarks (Himmi et al., 2023). The benchmarking paradigm is hence shifting towards adopting evaluation principles from other fields, such as non-parametric statistics and social choice theory (Brandt et al., 2016; Rofin et al., 2022). We use ordinal rankings instead of scores similar to ChatBot Arena. However, Arena systems use Elo-based scoring systems, well-established to be a poor metric (Boubdir et al., 2023), and our work confirms that. The pairwise variant of the Plackett-Luce model has been shown to have advantages both theoretically and empirically (Peyrard et al., 2021), allows us to inherit some of their theoretical properties like identifiability, sample-efficient convergence, provable robustness to irrelevant alternatives, non-dominance of outliers and empirical robustness properties across a wide range of real-world factors which affect ranking. In contrast, we do not aggregate over benchmarks, our primary proposal is avoid monolithic benchmarks and consider aggregation on a samplewise-level, needing to tackle incomplete and heterogeneous measurements.

**Efficient Evaluation.** As evaluation suites have grown in size, associated inference costs have also increased. Recent research has focused on creating compressed subsets of traditional benchmarks to address this issue (Varshney et al., 2022; Polo et al., 2024; Vivek et al., 2024; Zhao et al., 2024; Perlitz et al., 2024). Popular extensions include subsampling benchmarks to preserve correlations with an external source like ChatBot-Arena (Ni et al., 2024), or designing evolving sample-level benchmarks (Prabhu et al., 2024) similar in principle to our work. However, Prabhu et al. (2024) do not handle incomplete input matrices, which is necessary for aggregation over multiple timesteps and requires binary 0/1 evaluation metrics as input. We precisely address these limitations by showing efficient evaluation while accommodating for incomplete data and extending it to ordinal ranks in our work.

**Democratizing Evaluation.** Most standard image classification and retrieval benchmarks are collected from platforms like Flickr, which are predominantly Western-centric (Ananthram et al., 2024; Shankar et al., 2017). This has raised the important question: "Progress for whom?", with many seminal works showcasing large disparities in model performance on concepts (Nguyen et al., 2024; Hemmat et al., 2024), tasks (Hall et al., 2024; 2023b;a), and even input samples (Pouget et al., 2024; Sureddy et al., 2024; Gustafson et al., 2024) from the Global South. In response, works have developed benchmarks tailored to diverse cultures and demographics to include their voice in measuring progress (Pistilli et al., 2024; Pouget et al., 2024; Nguyen et al., 2024; Luccioni &

Rolnick, 2023). We take a different approach by creating flexible benchmarks where individuals, and contributing labs being able to add their own samples and preferences. During capability testing, users can select similar preferences, making $\infty$-benchmarks more inclusive than traditional test sets created by well-funded labs in wealthier countries.

# C DATASETS USED IN ∞-BENCHMARKS: FURTHER DETAILS

## C.1 ∞-LLMBENCH

| Dataset | Source | Task | Size | Metric | License |
|---|---|---|---|---|---|
| **Cardinal** | | | | | |
| LegalBench (Guha et al., 2024) | HELM | Legal | 1K | QEM | Unknown |
| MATH (Hendrycks et al., 2021b) | HELM | Maths | 1K | QEM | MIT |
| MedQA (Jin et al., 2021) | HELM | Medical | 1K | QEM | MIT |
| NarrativeQA (Kočiský et al., 2018) | HELM | Openbook QA | 1K | F1 | Apache-2.0 |
| NaturalQuestions (Kwiatkowski et al., 2019) | HELM | Search Engine Queries | 1K | F1 | CC BY-SA 3.0 |
| OpenbookQA (Mihaylov et al., 2018) | HELM | Openbook QA | 1K | EM | Apache-2.0 |
| WMT 2014 (Bojar et al., 2014) | HELM | Machine translation | 1K | BLEU | CC-BY-SA-4.0 |
| ARC (Clark et al., 2018) | Leaderboard | General QA | 1.1K | EM | CC-BY-SA-4.0 |
| HellaSwag (Zellers et al., 2019) | Leaderboard | Reasoning | 10K | EM | MIT |
| TruthfulQA (Lin et al., 2022) | Leaderboard | General QA | 817 | EM | Apache-2.0 |
| Winogrande (Sakaguchi et al., 2021) | Leaderboard | Reasoning | 1.2K | EM | Apache-2.0 |
| GSM8K (Cobbe et al., 2021) | HELM + Leaderboard | Maths | 1.3K | QEM | MIT |
| MMLU (Hendrycks et al., 2021a) | HELM + Leaderboard | General QA | 13.8K | EM | MIT |
| **Ordinal** | | | | | |
| Chatbot Arena Chiang et al. (2024) | Chatbot Arena | Pairwise Battles | 51K | - | CC BY 4.0 |

Table 3: **Datasets in ∞-LLMBench**: a diverse collection of benchmarks testing the abilities of LLMs in tasks such as law, medicine, mathematics, question answering, reasoning and instruction following as well as the performance of LLMs in pairwise battles.

## C.2 ∞-LMMBENCH

| Dataset | Source | Task | Size | Metric | License |
|---|---|---|---|---|---|
| **Cardinal** | | | | | |
| A-OKVQA (Schwenk et al., 2022) | VHELM | VQA | 7.2K | QEM | Apache-2.0 |
| Bingo (Cui et al., 2023) | VHELM | Bias+Hallucination | 886 | ROUGE | Unknown |
| Crossmodal-3600 (Thapliyal et al., 2022) | VHELM | Captioning | 1.5K | ROUGE | CC BY-SA 4.0 |
| Hateful Memes (Kiela et al., 2020) | VHELM | Hate Speech | 1K | QEM | Custom(Meta) |
| Mementos (Wang et al., 2024) | VHELM | Sequential Reasoning | 945 | GPT | CC-BY-SA-4.0 |
| MultipanelVQA (Fan et al., 2024) | VHELM | VQA | 200 | QEM | MIT |
| OODCV-VQA (Tu et al., 2023) | VHELM | VQA | 1K | QEM | CC-BY-NC-4.0 |
| PAIRS (Fraser & Kiritchenko, 2024) | VHELM | Bias+Hallucination | 508 | QEM | Unknown |
| Sketchy-VQA (Tu et al., 2023) | VHELM | VQA | 1K | QEM | CC-BY-NC-4.0 |
| AI2D (Kembhavi et al., 2016) | LMMs-Eval | Maths+Science | 3.09K | QEM | Apache-2.0 |
| IconQA (Lu et al., 2021) | LMMs-Eval | Docs and Infographics | 43K | ANLS | CC BY-SA 4.0 |
| InfoVQA (Mathew et al., 2022) | LMMs-Eval | Docs and Infographics | 6.1K | ANLS | Unknown |
| LLaVA-in-the-Wild (Liu et al., 2023a) | LMMs-Eval | Multi-disciplinary | 60 | GPT4 | Apache-2.0 |
| ChartQA (Masry et al., 2022) | LMMs-Eval | Docs and Infographics | 2.5K | QEM | GPL-3.0 |
| CMMMU (Zhang et al., 2024a) | LMMs-Eval | Multi-disciplinary | 900 | QEM | CC-BY-4.0 |
| DocVQA (Mathew et al., 2021) | LMMs-Eval | Docs and Infographics | 10.5K | ANLS | Unknown |
| MMBench (Liu et al., 2023b) | LMMs-Eval | Multi-disciplinary | 24K | GPT | Apache-2.0 |
| MMVET (Yu et al., 2024) | LMMs-Eval | Multi-disciplinary | 218 | GPT | Apache-2.0 |
| MP-DocVQA (Tito et al., 2023) | LMMs-Eval | Docs and Infographics | 5.2K | QEM | MIT |
| NoCaps (Agrawal et al., 2019) | LMMs-Eval | Captioning | 4.5K | ROUGE | MIT |
| OK-VQA (Marino et al., 2019) | LMMs-Eval | VQA | 5.1K | ANLS | Unknown |
| RefCOCO (Kazemzadeh et al., 2014; Mao et al., 2016) | LMMs-Eval | Captioning | 38K | ROUGE | Apache-2.0 |
| ScienceQA (Lu et al., 2022) | LMMs-Eval | Maths+Science | 12.6K | EM | CC BY-NC-SA 4.0 |
| TextCaps (Sidorov et al., 2020) | LMMs-Eval | Captioning | 3.2K | ROUGE | CC BY 4.0 |
| TextVQA (Singh et al., 2019) | LMMs-Eval | VQA | 5K | EM | CC BY 4.0 |
| COCO (Lin et al., 2014) | VHELM+LMMs-Eval | Captioning | 45.5K | ROUGE | CC-BY-4.0 |
| Flickr30k (Young et al., 2014) | VHELM+LMMs-Eval | Captioning | 31K | ROUGE | CC-0 Public Domain |
| GQA(Hudson & Manning, 2019) | VHELM+LMMs-Eval | Scene Understanding | 12.6K | QEM | CC-BY-4.0 |
| MathVista (Lu et al., 2024a) | VHELM+LMMs-Eval | Maths+Science | 1K | QEM/GPT4 | CC-BY-SA-4.0 |
| MME (Fu et al., 2023) | VHELM+LMMs-Eval | Multi-disciplinary | 2.4K | QEM/C+P | Unknown |
| MMMU (Yue et al., 2024) | VHELM+LMMs-Eval | Multi-disciplinary | 900 | QEM | CC BY-SA 4.0 |
| POPE (Li et al., 2023b) | VHELM+LMMs-Eval | Bias+Hallucination | 9K | QEM/EM | MIT |
| SEED-Bench (Li et al., 2023a; 2024a) | VHELM+LMMs-Eval | Multi-disciplinary | 42.5K | QEM/EM | Apache |
| VizWiz (Gurari et al., 2018) | VHELM+LMMs-Eval | VQA | 4.3K | QEM/EM | CC BY 4.0 |
| VQAv2 (Goyal et al., 2017) | VHELM+LMMs-Eval | VQA | 214K | QEM/EM | CC BY 4.0 |
| **Ordinal** | | | | | |
| Vision Arena (Lu et al., 2024b) | - | Pairwise Battles | 9K | - | MIT |
| LMMs-Eval(Prometheus2) (Kim et al., 2024) | - | Pairwise Battles | 610K | - | MIT |

Table 4: **Datasets in ∞-LMMBench**: a diverse collection of benchmarks testing the abilities of LLMs in tasks such as general VQA, Image Captioning, hate speech detection, bias and hallucination understanding, maths and science, documents and infographics, scene understanding and sequential reasoning as well as the performance of LMMs in pairwise battles.

# D    MODELS USED IN ∞-BENCHMARKS:FURTHER DETAILS

In this section, we provide a deeper insight into the models used in the creation of ∞-benchmarks. It is important to note that ∞-LLMBench and ∞-LMMBench have complementary characteristics: while ∞-LLMBench has fewer data samples $\mathcal{D}_k$, they are evaluated on more models $\mathcal{M}_k$, while ∞-LMMBench contains (significantly) more data samples but they are evaluated on less models.

## D.1    ∞-LLMBENCH: OPEN LLM LEADERBOARD

The Open LLM Leaderboard (Beeching et al., 2023) was created to track progress of LLMs in the open-source community by evaluating models on the same data samples and setup for more reproducible results and a trustworthy leaderboard where all open-sourced LLMs could be ranked.

However, due to the abundance of models found on the leaderboard and the lack of adequate documentation, and therefore reliability, of many of these models being evaluated, we rank the models based on the number of downloads, as a metric of adoption of these models by the community. We provide the total list of models as an artefact and list the top 100 models below:

1. 01-ai/Yi-34B-200K
2. AI-Sweden-Models/gpt-sw3-126m
3. BioMistral/BioMistral-7B
4. CohereForAI/c4ai-command-r-plus
5. CohereForAI/c4ai-command-r-v01
6. Deci/DeciLM-7B-instruct
7. EleutherAI/llemma_7b
8. EleutherAI/pythia-410m
9. Felladrin/Llama-160M-Chat-v1
10. Felladrin/Llama-68M-Chat-v1
11. FreedomIntelligence/AceGPT-7B
12. GritLM/GritLM-7B
13. Intel/neural-chat-7b-v3-1
14. JackFram/llama-160m
15. Nexusflow/NexusRaven-V2-13B
16. Nexusflow/Starling-LM-7B-beta
17. NousResearch/Hermes-2-Pro-Mistral-7B
18. NousResearch/Meta-Llama-3-8B-Instruct
19. NousResearch/Nous-Hermes-2-Mixtral-8x7B-DPO
20. NousResearch/Nous-Hermes-2-SOLAR-10.7B
21. NousResearch/Nous-Hermes-2-Yi-34B
22. OpenPipe/mistral-ft-optimized-1227
23. Qwen/Qwen1.5-0.5B
24. Qwen/Qwen1.5-0.5B-Chat
25. Qwen/Qwen1.5-1.8B
26. Qwen/Qwen1.5-1.8B-Chat
27. Qwen/Qwen1.5-110B-Chat
28. Qwen/Qwen1.5-14B
29. Qwen/Qwen1.5-14B-Chat
30. Qwen/Qwen1.5-32B-Chat
31. Qwen/Qwen1.5-4B
32. Qwen/Qwen1.5-4B-Chat
33. Qwen/Qwen1.5-72B-Chat
34. Qwen/Qwen1.5-7B
35. Qwen/Qwen1.5-7B-Chat
36. SeaLLMs/SeaLLM-7B-v2
37. TinyLlama/TinyLlama-1.1B-Chat-v1.0
38. TinyLlama/TinyLlama-1.1B-intermediate-step-1431k-3T
39. VAGOsolutions/SauerkrautLM-Mixtral-8x7B-Instruct
40. abhishekchohan/mistral-7B-forest-dpo
41. ahxt/LiteLlama-460M-1T
42. ai-forever/mGPT
43. alignment-handbook/zephyr-7b-sft-full
44. augmxnt/shisa-gamma-7b-v1
45. bigcode/starcoder2-15b
46. bigcode/starcoder2-3b
47. bigcode/starcoder2-7b
48. cloudyu/Mixtral_7Bx4_MOE_24B
49. codellama/CodeLlama-70b-Instruct-hf
50. cognitivecomputations/dolphin-2.2.1-mistral-7b
51. cognitivecomputations/dolphin-2.6-mistral-7b-dpo
52. cognitivecomputations/dolphin-2.9-llama3-8b
53. daekeun-ml/phi-2-ko-v0.1
54. deepseek-ai/deepseek-coder-1.3b-instruct
55. deepseek-ai/deepseek-coder-6.7b-base
56. deepseek-ai/deepseek-coder-6.7b-instruct
57. deepseek-ai/deepseek-coder-7b-instruct-v1.5
58. deepseek-ai/deepseek-math-7b-base
59. deepseek-ai/deepseek-math-7b-instruct
60. deepseek-ai/deepseek-math-7b-rl
61. google/codegemma-7b-it
62. google/gemma-1.1-7b-it
63. google/gemma-2b
64. google/gemma-2b-it
65. google/gemma-7b
66. google/gemma-7b-it
67. google/recurrentgemma-2b-it
68. h2oai/h2o-danube2-1.8b-chat
69. hfl/chinese-alpaca-2-13b
70. ibm/merlinite-7b
71. meta-llama/Meta-Llama-3-70B
72. meta-llama/Meta-Llama-3-70B-Instruct
73. meta-llama/Meta-Llama-3-8B
74. meta-llama/Meta-Llama-3-8B-Instruct
75. meta-math/MetaMath-Mistral-7B
76. microsoft/Orca-2-7b
77. microsoft/phi-2
78. mistral-community/Mistral-7B-v0.2
79. mistral-community/Mixtral-8x22B-v0.1
80. mistralai/Mistral-7B-Instruct-v0.2
81. mistralai/Mixtral-8x22B-Instruct-v0.1
82. mistralai/Mixtral-8x7B-Instruct-v0.1
83. mistralai/Mixtral-8x7B-v0.1
84. openai-community/gpt2
85. openai-community/gpt2-large
86. openchat/openchat-3.5-0106

87. openchat/openchat-3.5-1210
88. openchat/openchat_3.5
89. sarvamai/OpenHathi-7B-Hi-v0.1-Base
90. speakleash/Bielik-7B-Instruct-v0.1
91. speakleash/Bielik-7B-v0.1
92. stabilityai/stablelm-2-1_6b
93. stabilityai/stablelm-2-zephyr-1_6b

94. stabilityai/stablelm-zephyr-3b
95. teknium/OpenHermes-2.5-Mistral-7B
96. tokyotech-llm/Swallow-70b-instruct-hf
97. upstage/SOLAR-10.7B-Instruct-v1.0
98. upstage/SOLAR-10.7B-v1.0
99. wenbopan/Faro-Yi-9B
100. yanolja/EEVE-Korean-Instruct-10.8B-v1.0

## D.2 ∞-LLMBENCH: HELM

Similar to the Open LLM Leaderboard, the goal of HELM was to provide a uniform evaluation of language models over a vast set of data samples (termed as `scenarios` in Liang et al. (2023)). HELM, however, has a broader scope of models used for evaluation, employing open, limited-access, and closed models. All models currently used in ∞-LLMBench is listed below:

1. 01-ai_yi-34b
2. 01-ai_yi-6b
3. 01-ai_yi-large-preview
4. ai21_j2-grande
5. ai21_j2-jumbo
6. ai21_jamba-1.5-large
7. ai21_jamba-1.5-mini
8. ai21_jamba-instruct
9. AlephAlpha_luminous-base
10. AlephAlpha_luminous-extended
11. AlephAlpha_luminous-supreme
12. allenai_olmo-7b
13. anthropic_claude-2.0
14. anthropic_claude-2.1
15. anthropic_claude-3-5-sonnet-20240620
16. anthropic_claude-3-haiku-20240307
17. anthropic_claude-3-opus-20240229
18. anthropic_claude-3-sonnet-20240229
19. anthropic_claude-instant-1.2
20. anthropic_claude-instant-v1
21. anthropic_claude-v1.3
22. cohere_command
23. cohere_command-light
24. cohere_command-r
25. cohere_command-r-plus
26. databricks_dbrx-instruct
27. deepseek-ai_deepseek-llm-67b-chat
28. google_gemini-1.0-pro-001
29. google_gemini-1.0-pro-002
30. google_gemini-1.5-flash-001
31. google_gemini-1.5-pro-001
32. google_gemini-1.5-pro-preview-0409
33. google_gemma-2-9b-it
34. google_gemma-2-27b-it
35. google_gemma-7b
36. google_text-bison@001
37. google_text-unicorn@001
38. meta_llama-2-7b
39. meta_llama-2-13b
40. meta_llama-2-70b
41. meta_llama-3-8b
42. meta_llama-3-70b
43. meta_llama-3.1-8b-instruct-turbo
44. meta_llama-3.1-70b-instruct-turbo
45. meta_llama-3.1-405b-instruct-turbo
46. meta_llama-65b
47. microsoft_phi-2
48. microsoft_phi-3-medium-4k-instruct
49. mistralai_mistral-7b-instruct-v0.3
50. mistralai_mistral-7b-v0.1
51. mistralai_mistral-large-2402
52. mistralai_mistral-large-2407
53. mistralai_mistral-medium-2312
54. mistralai_mistral-small-2402
55. mistralai_mixtral-8x7b-32kseqlen
56. mistralai_mixtral-8x22b
57. mistralai_open-mistral-nemo-2407
58. nvidia_nemotron-4-340b-instruct
59. openai_gpt-3.5-turbo-0613
60. openai_gpt-4-0613
61. openai_gpt-4-1106-preview
62. openai_gpt-4-turbo-2024-04-09
63. openai_gpt-4o-2024-05-13
64. openai_gpt-4o-mini-2024-07-18
65. openai_text-davinci-002
66. openai_text-davinci-003
67. qwen_qwen1.5-7b
68. qwen_qwen1.5-14b
69. qwen_qwen1.5-32b
70. qwen_qwen1.5-72b
71. qwen_qwen1.5-110b-chat
72. qwen_qwen2-72b-instruct
73. snowflake_snowflake-arctic-instruct
74. tiiuae_falcon-7b
75. tiiuae_falcon-40b
76. writer_palmyra-x-004
77. writer_palmyra-x-v2
78. writer_palmyra-x-v3

### D.3   ∞-LMMBENCH: LMMS-EVAL

LMMs-Eval is the first comprehensive large-scale evaluation benchmark for Large Multimodal models, meant "to promote transparent and reproducible evaluations" (Zhang et al., 2024b). The models supported by LMMs-Eval are primarily open-sourced and the full list of currently used models are listed below:

1. idefics2-8b
2. internlm-xcomposer2-4khd-7b
3. instructblip-vicuna-7b
4. instructblip-vicuna-13b
5. internVL-Chat-V1-5
6. llava-13b
7. llava-1.6-13b
8. llava-1.6-34b
9. llava-1.6-mistral-7b
10. llava-1.6-vicuna-13b
11. llava-1.6-vicuna-7b
12. llava-7b
13. llava-next-72b
14. qwen_vl_chat

### D.4   ∞-LMMBENCH: VHELM

Finally, ∞-LMMBench comprises VHELM, an extension of HELM for Vision-Language models. The models currently used by us, spanning open, limited-access, and closed models, are as follows:

1. anthropic_claude_3_haiku_20240307
2. anthropic_claude_3_opus_20240229
3. anthropic_claude_3_sonnet_20240229
4. google_gemini_1.0_pro_vision_001
5. google_gemini_1.5_pro_preview_0409
6. google_gemini_pro_vision
7. google_paligemma_3b_mix_448
8. huggingfacem4_idefics2_8b
9. huggingfacem4_idefics_80b
10. huggingfacem4_idefics_80b_instruct
11. huggingfacem4_idefics_9b
12. huggingfacem4_idefics_9b_instruct
13. llava_1.6_mistral_7b
14. llava_1.6_vicuna_13b
15. llava_1.6_vicuna_7b
16. microsoft_llava_1.5_13b_hf
17. microsoft_llava_1.5_7b_hf
18. mistralai_bakllava_v1_hf
19. openai_gpt_4_1106_vision_preview
20. openai_gpt_4_vision_preview
21. openai_gpt_4o_2024_05_13
22. openflamingo_openflamingo_9b_vitl_mpt7b
23. qwen_qwen_vl
24. qwen_qwen_vl_chat
25. writer_palmyra_vision_003

# E CAPABILITY TESTING ACROSS ARBITRARY QUERIES

## E.1 QUERIES: LIST AND QUANTITATIVE RESULTS

| Concept | $\infty$-LLMBench AP | $\infty$-LMMBench AP |
|---|---|---|
| Common Queries | | |
| apple ipad | 0.7435 | 0.1985 |
| architecture | 0.7683 | 0.8981 |
| beach | 0.7152 | 0.5698 |
| biochemistry | 0.9778 | 0.7303 |
| boat | 0.7728 | 0.8829 |
| botany | 0.9876 | 0.7556 |
| bus | 0.9035 | 0.9739 |
| car | 0.9140 | 0.8477 |
| cell(biology) | 0.9937 | 0.5075 |
| china tourism | 0.6392 | 1.0000 |
| cigarette ads | 0.7249 | 0.6590 |
| coffee maker | 0.8426 | 0.4057 |
| components of a bridge | 0.9222 | 0.5865 |
| decomposition of benzene(organic chemistry) | 0.6745 | 0.7623 |
| epidemiology | 0.9316 | 0.7991 |
| feminist theory | 0.8566 | 0.5138 |
| kirchoffs law(electrical engineering) | 0.6572 | 0.4824 |
| food chain | 0.5405 | 1.0000 |
| game of football | 0.8221 | 1.0000 |
| german shepherd (dog) | 0.9359 | 0.3078 |
| gothic style (architecture) | 0.7829 | 1.0000 |
| literary classics | 0.9869 | 1.0000 |
| macroeconomics | 1.0000 | 0.9570 |
| makeup | 1.0000 | 0.2247 |
| microwave oven | 0.7979 | 1.0000 |
| neuroscience components | 0.9844 | 0.2854 |
| pasta | 0.5678 | 0.2142 |
| perfume | 0.5996 | 0.6355 |
| photosynthesis | 0.9848 | 0.3665 |
| plants | 1.0000 | 0.6488 |
| political diplomacy | 0.9529 | 0.9561 |
| python code | 0.8850 | 0.9444 |
| renaissance painting | 0.9270 | 0.9799 |
| shareholder report | 1.0000 | 0.8317 |
| sheet music | 0.8322 | 0.9750 |
| solar cell battery | 0.8853 | 0.8082 |
| thermodynamics | 0.9567 | 0.8852 |
| united states of america | 0.8096 | 0.8642 |
| vaccines | 0.8572 | 0.3411 |
| volanic eruption | 0.7905 | 0.9229 |
| Queries testing Visual Capabilities | | |
| bike leaning against wall | - | 0.8271 |
| child playing baseball | - | 0.9638 |
| coriolis effect | - | 0.7063 |
| dijkstras shortest path algorithm | - | 0.9135 |
| empty bridge overlooking the sea | - | 0.5934 |
| judo wrestling | - | 0.6092 |
| man in a suit | - | 0.5611 |
| musical concert | - | 0.9879 |
| sine wave | - | 0.4232 |
| woman holding an umbrella | - | 0.8821 |

Table 5: Aggregate Average Precision(AP) for $\infty$-LLMBench and $\infty$-LMMBench concepts.

## E.2 QUALITATIVE RESULTS

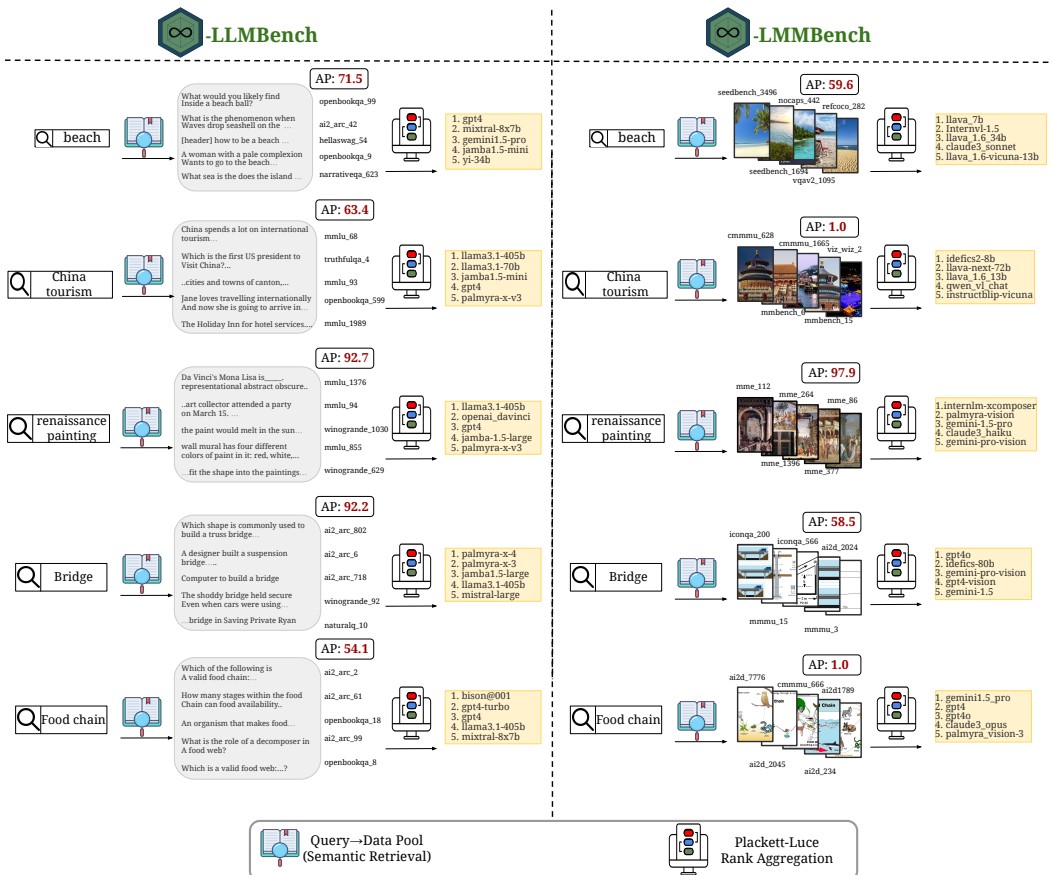

Figure 7: Additional qualitative analysis for ∞-benchmark's capability probing for selected queries.

