# OpenReview forum: "Democratizing Evaluation with Infinity-Benchmarks: Sample-Level Heterogeneous Testing Over Arbitrary Capabilities"
_ICLR.cc/2025/Conference — ICLR 2025 Conference Withdrawn Submission_

### Official Review · Reviewer_aKaM · 2024-11-01

**Soundness:** 3
**Presentation:** 1
**Contribution:** 2
**Rating:** 3
**Confidence:** 4

**Summary:**

This paper broadly makes two different contributions:

1. The paper proposes treating all (per-sample) evaluation scores of language models as (per-sample) rankings and then demonstrates that such sample-leve rankings can be used to rank such models using the Plackett-Luce model.
2. The authors also argue that, since all per-sample ranking scores can be generally aggregated, one can dynamically carve up existing benchmarks or construct new benchmarks on the fly by segmenting the data pool into relevant subsets and then aggregating ranking scores from that subset.

**Strengths:**

- The grammar, styling, attention to detail, etc. is extremely high quality
- Figure 1is very well done and communicates the paper’s contributions well
- Table 1 is a reasonable way to assess different ranking mechanisms
- Figure 3 is compelling

**Weaknesses:**

- Overall, this paper relies on a critical assumption: that model rankings are all the field cares about. For instance, in Section 2.2 Line 244, the manuscript states: “For practitioners, the critical concern is whether the top models are ranked correctly.” I think this assumption is generally false. From personal experience, practitioners care very much about the magnitude/size of gaps between models’ capabilities. For instance, when OpenAI’s o1 came out, it was ranked #1 in AIME, and its performance was significantly better than that of any competitor. When GDM announced a 1M and 10M context length, that context length wasn’t merely #1 - such a context length was 1-2 orders of magnitude longer than any other available model at the time. Similarly, for companies or startups or organizations using these models, if Model B is epsilon-better than Model A, but Model A is already implemented and trustworthy and cheap, switching to Model B isn’t worth the effort and potential risks. Cost is an especially important factor; a model that is epsilon-better likely isn’t worth a 10x or 100x inference costs.  Consequently, I find this paper’s contributions limited because I disagree with its premise that ordinal rankings of models are the important/meaningful signal.

- In terms of writing, this paper is difficult to follow. By Section 2.2, I was lost. I then discovered that the methodology that this paper contributes cannot be found in the main text and is instead detailed in Appendix A (I also cannot find a reference to Appendix A in the main text, although this might be my own inability). I feel like Appendix A is a significant pillar of this paper and should thus be included prominently in the main text.

- Section 2.1: This is the first substantive section of the paper, and it is titled “Why This Works.” Up to this point in the paper, the reader does not know what exactly “this” is and has no evidence that whatever “this” is does work. Consequently, it seems premature at this point in the manuscript to explain why some unknown method achieves some unknown performance.

- Section 2.1: I don’t know what “P1, P2, P3…” refer to. Please state what “P” is an abbreviation of.

- Section 2.1: The text in each paragraph appears to be a literature review of the Plackett-Luce and the properties that accompany it. In this section, I am not able to identify what these authors are contributing in this section of text.

- Appendix A: Minor: The method assumes that sample-level rankings are available, which is oftentimes not true. Oftentimes, model creators release only aggregate metrics for an entire benchmark (e.g., 5-Shot Accuracy on MMLU).

- Appendix A: Line 1175-1176 states, “In practice, ordinal measurements can paradoxically outperform cardinal ones despite the inherent information loss.” While I buy that ordinal measurements can outperform cardinal ones, whether they actually do so in a particular setting remains to be proven, and it is incumbent upon the researchers to demonstrate this.

- Given that this paper relies heavily on the Plackett-Luce model, the authors absolutely must state/summarize what it is. I was not intimately familiar and had to go educate myself.

- Figure 2: Echoing my first point about how ordinal ranks are not sufficient, I have no sense of whether these permutations of rankings are minor or major.

- Lines 304 and 310: How does insight 2 not contradict insight 3? Insight 3 states that Random Sampling matches Informative Sampling, but Insight 2 (and the subsection title “ACTIVE SAMPLING IMPROVES DATA AGGREGATION EFFICIENCY”) seem to contradict this. It seems like the claims of this section and of Figure 4 (bottom) is self-contradictory.

- Figure 5 is aesthetically nice but lacking in substance

**Questions:**

- In Section 2.1, what does “P” stand for in “P1.”, “P2.”, etc.?
- Table 1: Why are the Kendall taus with LLMs-Eval and VHELM comparatively low for your method? How is LLMs–Eval scored such that the Plackett-Luce only has a correlation of 0.67?
- Line 236: “Our method preserves the ranking of the top-10 models.” I might be misreading the figure, but it seems that your method reorders the ground truth rankings? If so, how does your method preserve the ranking of the top 10 models.

---

### Official Review · Reviewer_knSj · 2024-11-03

**Soundness:** 3
**Presentation:** 2
**Contribution:** 2
**Rating:** 5
**Confidence:** 3

**Summary:**

The paper proposes aggregating datasets from existing benchmarks to allow on-the-fly construction of customized benchmarks. To accomplish this goal, the authors introduce a ranking-based model that boasts high sample efficiency. The authors focus their attention on the problems of 1) how to combine different metrics (which the authors address with a ranking-based approach) and 2) how to compare models evaluated on different data examples (the authors lean on theoretical properties of their proposed methodology, guaranteeing that models can be ranked if each pair of models is connected by a directed path of pairwise rankings). The authors study the sample efficiency of their methodology and investigate the possibility of making benchmarks less resource-intensive by removing low-signal data examples. Finally, the authors illustrate the construction of custom benchmarks for queries such as "neuroscience" or "perfume".

**Strengths:**

The proposed infinity-benchmarks pose an interesting and important question. Can we dynamically create benchmarks on the fly to evaluate foundation model performance more flexibly? This approach is interesting because custom benchmarks can give fine-grained insights into model behavior, which may be obscured when we focus on average performance trends on the most popular benchmarks. In addition, aggregating data from a pool of different data sets may reduce the impact of statistical artifacts/ideosyncrasies present in individual data sets.

To implement this idea, the authors propose the Plackett-Luce model for computing model performance rankings, and they describe the theoretical conditions under which the model is identifiable. The resulting ranking methodology is highly sample efficient, which is a strong advantage given the cost and limited scalability of hiring human data labelers, especially on more specialized subjects requiring subject matter expertise.

**Weaknesses:**

The paper proposes heterogeneity and incompleteness – implementation issues – as the main obstacles hindering usage of dynamic benchmarks. However, there are broader questions about the usefulness and desirability of custom benchmarks, which are largely unaddressed:

1) Do dynamically constructed benchmarks actually measure the capabilities of interest? The authors partially address this question in Section 3.2. But there are important unaddressed questions, such as the robustness of custom benchmarks' informativeness to imperfect retrieval of data examples. It would also be interesting to explore how specific the custom benchmarks can be, and whether more specific benchmarks (e.g., "Kirchoff's Law") face greater issues with retrieval precision compared to broader benchmarks (e.g., "electricity and magnetism"). Overall, I would like to see a more detailed investigation on whether dynamically constructed benchmarks actually measure capabilities of interest.

2) There are consequences to adopting dynamic benchmarks within the research community. Would dynamic benchmarks incentivize researchers to invent new benchmarks for their papers, allowing them to claim SOTA on narrow tasks? Would researchers try to create many different custom benchmarks until they find one on which their new method performs well?

3) The re-use of data examples from the same pool results in correlations between different custom benchmarks. Phrased in a different way, a single data example can affect many different custom benchmarks. When dynamic benchmarks are used to report model performance across different research papers, how can we transparently present the correlations and dependencies between different custom benchmarks?

In addition to these big picture concerns, I have more concrete questions about the work, for example about the soundness of the ranking evaluation with respect to ground truth. Please see the Questions.

**Questions:**

1. You are measuring the performance of your model ranking methodology by comparing with rankings of existing benchmarks (see Table 1). These existing benchmarks consist of different subtasks and must tackle the problem of aggregating performance data from different subtasks, just like your method aims to do. For example, the HELM leaderboard ranks models by mean win rate against other models. Hence, it is unclear if these rankings can be considered “ground truth” rather than just different ways of computing model rankings. For example, what if HELM actually used your method to aggregate performance from its constituent tasks? Then your method would trivially reach perfect agreement. Does taking the rankings of existing benchmarks as “ground truth” make sense?

2. Your work focuses on implementing custom benchmarks via ranking. However, beyond ranking, it is useful to have absolute performance numbers. For example, if one model reaches 99% accuracy and another 98.9%, it may not matter in practice which model to use. Do you see any possibilities for dynamic benchmarks to reveal when the differences in ranking between models are significant?

3. In Section 3.2.1, the average precision (AP) for retrieving data examples ranges from 28.5% to 100%. Erroneously retrieved data examples may affect the overall ranking, reducing fidelity to the user’s benchmarking goal. In the experiment you conducted, does filtering out the erroneously retrieved data examples change the ordering of the top-5 models?

4. You show that random selection of data examples and informative selection of data examples result in similar performance. How does this align with your discussion of low-signal data examples? For example, eyeballing Figure 4 suggests that on the Open LLM Leaderboard, perhaps 40% of data examples are low-signal (all or none of the models answer correctly). This suggests that informative sampling – focusing on data examples where different models perform differently – should be able to get away with using only 60% as many data examples as random sampling. Could you highlight this benefit of informative sampling in your paper, or else explain why it does not materialize?

---

### Official Review · Reviewer_SrCU · 2024-11-03

**Soundness:** 2
**Presentation:** 2
**Contribution:** 3
**Rating:** 3
**Confidence:** 3

**Summary:**

The paper makes two contributions to LLM capability evaluations. First, it proposes to use Plackett-Luce model to rank LLMs and has shown that it is more robust to missing values in evaluation. Second, it proposes personalized evaluation where the user can submit queries that represent capabilities they are interested in, and the evaluation will focus on the subset of samples that best match the queries. This has shown to reveal different LLM rankings depending on the queries.

**Strengths:**

The idea of having personalized evaluation makes sense and it can be very useful to practitioners who work on different fields.

**Weaknesses:**

I am not very familiar with social choice theory, but it still seems to me that the paper is not very written in Section 2. Specifically,
- Can the authors explain exactly how Plackett-Luce model obtains a predicted ranking from observations of performance of LLMs on samples?
- Can the authors explain why no metric achieves 1.0 Kendall Tau correlation in Table 1 when full data is used to evaluate model rankings? Where does the ground truth model ranking come from in this case?

The paper also makes several observations that is too trivial in my opinion.
 - In Section 2.3, the paper mentions that for a dataset there exists many easy and hard problems where either all models get right or no models get right. Therefore, when performing evaluation, we can be efficient and just use the problems that come from the central difficulty bin (Figure 4). Isn't this quite obvious? Also, in order to know what problems are in the central bin, we need to already evaluate models on these problems. It seems to defeat the purpose of efficient evaluation. Additionally, whether a problem belongs to central bin is dynamic depending on the change of model capabilities.

- In Section 3.2, while I like the concept of personalized evaluation, I still believe this is a straightforward idea without much technical contribution. It can be thought as a retrieval + evaluation problem where depending on what concept a user wants to evaluate, we can retrieve the questions from that concept and then evaluate. People have also explored this in a more high level where to evaluate the mathematical reasoning capabilities, datasets like GSM8K, MATH will be used to evaluate.

The paper should also cite a few related work such as [1,2, 3] about efficient evaluation using multi-arm bandits since this is also mentioned in the paper.

[1] Shi, Chengshuai, Kun Yang, Zihan Chen, Jundong Li, Jing Yang, and Cong Shen. "Efficient prompt optimization through the lens of best arm identification." arXiv preprint arXiv:2402.09723 (2024).

[2] Zhou, Jin Peng, Christian K. Belardi, Ruihan Wu, Travis Zhang, Carla P. Gomes, Wen Sun, and Kilian Q. Weinberger. "On Speeding Up Language Model Evaluation." arXiv preprint arXiv:2407.06172 (2024).

**Questions:**

Please see the Weakness section above.

---

### Official Review · Reviewer_MH8B · 2024-11-04

**Soundness:** 2
**Presentation:** 2
**Contribution:** 2
**Rating:** 3
**Confidence:** 3

**Summary:**

This paper proposes ∞-benchmarks, an evaluation paradigm for ranking and understanding foundation models.

**Strengths:**

As foundation models are developed on top of huge amounts of data, it is a timely and important question to study how to evaluate and understand these models.

**Weaknesses:**

While I appreciate the importance of evaluating foundation models, I found it difficult to understand the proposed ∞-benchmarks and thus the main contribution of this paper. In particular, what is ∞-benchmarks? Is it a new dataset, an evaluation pipeline, or a model ranking tool?

Half of the paper's technical content is about how the proposed ranking method outperforms the other methods (page 3-6). The authors use the correlation between the ground-truth ranking and the learned ranking as the metric for the comparison. What is the ground-truth ranking? Why are the ground-truth rankings independent of the ranking methods? As far as I understand, rankings are somewhat subjective, and the ground-truth sometimes are determined by the ranking methods directly. For example, Elo-score simply defines the ground-truth rankings by (sufficiently) many battles between all players/foundation models.

The other half of the paper's technical meat presents LLMBench and LMMBench. While the analysis on which models are performative on which queries is insightful, I am not sure what the technical contribution is. Is it simply merging many existing datasets into one, or is there anything I am missing here?

Overall, I find it is very hard to tell what the contribution is given the current form of this paper.

**Questions:**

See my questions above.

---

### Author Response · Authors · 2024-11-24
**Author Rebuttal**

We appreciate the reviewers' feedback and would like to address the most commonly raised issues in this joint response.

1) **What are ∞-benchmarks?**

Our main technical contribution is a method for aggregating model evaluations across incompatible metrics and diverse data sources. We achieve this by converting all measurements from the cardinal form (individual measurements such as accuracy or BLEU score) to the ordinal form (pairwise comparisons between two or more models) and applying a random utility model based on the Plackett-Luce framework. In essence, we demonstrate a principled way to produce a unified ranking and model scores from heterogeneous and incomplete measurements.

This approach enables decentralised model benchmarking that supports lifelong, continuously updated, and ad-hoc sample-level evaluation. We demonstrate its viability by integrating information between HELM and Open LLM Leaderboard, as well as VHELM and LMMs-Eval.

2) **The description of the Plackett-Luce model is not clear.**

We acknowledge this feedback and agree that our description of the random utility model could be improved. We will move relevant details from the appendix to the main paper to enhance clarity.

3) **Do practitioners care about rankings?**

The emergence of projects like HELM or the Open LLM Leaderboard show that the community does care about aggregating individual benchmark scores into rankings. At the same time, the popularity of platforms like Chatbot Arena demonstrates that scores derived from pairwise comparisons are a viable method for measuring model performance. While ELO or Bradley-Terry scores are relative, they effectively convey information about the magnitude of gaps between model capabilities. Similarly, scores from the Plackett-Luce model help practitioners gauge when differences in model rankings are significant. We will revise our paper to emphasise that our method provides both rankings and scores. Additionally, we will analyse how well these scores correlate with absolute values of individual metrics on homogeneous datasets.

4) **What is the ground truth in Section 3?**

We acknowledge that Section 3 needs greater clarity regarding the ground truth used to calculate ranking correlations. While our method is the first to enable aggregation of evaluations across benchmarks, Section 3 focuses on demonstrating its ability to recover score-based rankings within uniform benchmarks. We therefore compare all ranking methods against score-based rankings within each benchmark, obtained by normalising numerical metrics, averaging them per benchmark, and sorting models by their aggregated scores.

5) **The informative sampling method is inefficient.**

We apologise for any confusion regarding this point. Our intention was to demonstrate that the random strategy performs as well as the informative one, thus eliminating the need for the complete evaluation required by the latter. We will edit this section in future versions of the manuscript.

6) **Are personalised benchmarks robust?**

The quality and relevance of retrieved data samples depends on both the query precision and the size of the data pool. In our examples, we restrict ourselves to simple queries over the combined data pools of HELM and LLM Leaderboard for LLMs, and VHELM and LMMs-Eval for VLMs. However, our implementation supports both straightforward semantic search and structured, compositional filters. Our vision is to develop a querying mechanism over an open, distributed, continuously expanding, and potentially crowdsourced benchmark with a data pool large enough to provide comprehensive coverage of popular concepts.

7) **Do dynamically constructed benchmarks measure the capabilities of interest?**

With our definition of capability probing, we state that it is possible to query any arbitrary concept and find representative samples in ∞-benchmarks. And while our concept pool is a proof-of-concept (as highlighted in the submission), we took into consideration the hierarchy of concepts and provide quantitative analysis of how accurate and representative the retrieved samples are. For example, we query architecture and gothic architecture, where the latter is a more fine-grained query of the former. Based on the review and filtering of mismatched samples by expert annotators, we observe an average precision (AP) of 77% and 90% for the concept ‘architecture’ on ∞-LLMBench and ∞-LMMBench respectively and an AP of 79% and 100% for ‘gothic architecture’. We aim to address the removal of irrelevant retrieved samples by setting a similarity threshold for the similarity score between the query and ∞-benchmarks sample embeddings.

We sincerely thank all reviewers for their valuable insights. We will incorporate this feedback in future submissions to enhance the paper's clarity and coherence.

---

### Note · Authors · 2024-11-25

I have read and agree with the venue's withdrawal policy on behalf of myself and my co-authors.